# Medical Experts’ Agreement on Risk Assessment Based on All Possible Combinations of the COVID-19 Predictors—A Novel Approach for Public Health Screening and Surveillance

**DOI:** 10.3390/ijerph192416601

**Published:** 2022-12-10

**Authors:** Mohd Salami Ibrahim, Nyi Nyi Naing, Aniza Abd Aziz, Mokhairi Makhtar, Harmy Mohamed Yusoff, Nor Kamaruzaman Esa, Nor Iza A Rahman, Myat Moe Thwe Aung, San San Oo, Samhani Ismail, Ras Azira Ramli

**Affiliations:** 1Faculty of Medicine, Medical Campus, Universiti Sultan Zainal Abidin, Kuala Terengganu 20400, Terengganu, Malaysia; 2Faculty of Informatics and Computation, Gong Badak Campus, Universiti Sultan Zainal Abidin, Kuala Terengganu 20300, Terengganu, Malaysia

**Keywords:** agreement, combinations, dashboard-based rating, novel method, public health screening, public health surveillance, risk assessment

## Abstract

During the initial phase of the coronavirus disease 2019 (COVID-19) pandemic, there was a critical need to create a valid and reliable screening and surveillance for university staff and students. Consequently, 11 medical experts participated in this cross-sectional study to judge three risk categories of either low, medium, or high, for all 1536 possible combinations of 11 key COVID-19 predictors. The independent experts’ judgement on each combination was recorded via a novel dashboard-based rating method which presented combinations of these predictors in a dynamic display within Microsoft Excel. The validated instrument also incorporated an innovative algorithm-derived deduction for efficient rating tasks. The results of the study revealed an ordinal-weighted agreement coefficient of 0.81 (0.79 to 0.82, *p*-value < 0.001) that reached a substantial class of inferential benchmarking. Meanwhile, on average, the novel algorithm eliminated 76.0% of rating tasks by deducing risk categories based on experts’ ratings for prior combinations. As a result, this study reported a valid, complete, practical, and efficient method for COVID-19 health screening via a reliable combinatorial-based experts’ judgement. The new method to risk assessment may also prove applicable for wider fields of practice whenever a high-stakes decision-making relies on experts’ agreement on combinations of important criteria.

## 1. Introduction

Risk assessment based on various combinations of predictors is one of the commonest practices in medicine and healthcare. For example, evidence has been established to use varying mixtures of major and minor criteria to diagnose endocarditis [1], Wells score to predict pulmonary embolism in the emergency setting [2], and combinations of CURB-65 criteria to screen patients with pneumonia who will be needing hospital admission [3]. In May 2020, medical researchers from Universiti Sultan Zainal Abidin (UniSZA) adopted the combinatorial-based risk assessment to update the COVID-19 health screening and self-evaluation (CHaSe). CHaSe is a screening system accessible via mobile apps and web browsers. CHaSe is also extended as a surveillance system for active case detection when all UniSZA staff and students have been asked to complete daily risk assessments using CHaSe prediction model for every working day [4].

The original CHaSe risk prediction model consisted of 11 predictors, which were selected and validated from all known predictors of COVID-19 at that time via the Six Sigma approach [4,5]. CHaSe screening only approved access to the campus grounds for staff and students under the low-risk category. Respondents identified as medium and high risk, on the other hand, were being guided to access timely and appropriate further clinical assessments. Therefore, updating risk screening based on reliable experts’ judgement on various combinations of predictors has emerged as one of the priorities for the university health policy.

Reliable experts’ judgement as the source of validity is widely recognized in many fields and practices. These fields include institutional audit and quality appraisal [6], research methodologies such as content validity and face validity of instrument development [7], risk assessment of credit score in the financial sector [8], and the standard-setting method in the educational field to determine the passing mark of a high-stakes examination [9]. Hence, reliable experts’ judgement brought relevance, confidence, and stakeholders’ buy-in on the validity of the updated risk prediction model and the ensuing action recommendations.

Unfortunately, there is a lack of an established and practical method to determine the extent of agreement among medical experts on risk assessment based on combinations of risk factors. Our initial attempt was to conduct a pilot study by inviting six medical experts from the Faculty of Medicine into a seminar room. Each medical expert was then asked to indicate the rating of risk categories whenever one of the researchers displayed various combinations of key predictors of COVID-19. They indicated rating by raising the cardboard showing “low”, “medium”, or “high”. This exercise elucidated four key lessons. First, experts’ ratings were considerably varied. Second, independent experts’ rating of all possible combinations of predictors was necessary to establish the extent of experts’ agreement statistically. Third, the complete rating on all possible combinations of the predictors was also required to determine the level of screening stringency acceptable by the medical experts. Finally, the cognitive overload associated with the task of rating all possible combinations was a crucial limitation.

The final lesson was the key message echoed by many experts who enquired if there was a way for them not to repeat many ratings which could readily be deduced from prior ratings. For example, suppose an expert had already rated respondents with dry cough, sore throat, and fever under the high-risk category. Then all other combinations which contain these three predictors, regardless of other predictors, could also be readily deduced as high-risk. The feedback signified that an innovation is needed to circumvent the overwhelming burden of tasks to judge risk categories for all possible combinations of predictors. They also showed that conventional questionnaire-type data collection was practically not possible.

Several possible alternatives to risk assessment for public health screening had been reported in the literature. For example, to prevent deaths attributable to colorectal cancer, Ladabaum, Dominitz [10] advocated improved access to screening modalities such as guaiac-based fecal occult blood tests (gFOBTs), flexible sigmoidoscopy, multi-target stool DNA (mt-sDNA) test, and computed tomography colonography (CTC). Their experts’ appraisal of evidence appeared to focus on the sensitivity, specificity, and predictive values of these tests for decision-making on screening [10]. In comparison, the systematic review of the literature by Saulle, Sinopoli [11] assessed five studies on breast cancer, five studies on cervical cancer, and three studies on chronic and degenerative diseases which employed PRECEDE-PROCEED risk assessment for public health screening. The “Precede” (Predisposing Reinforcing and Enabling Constructs in Educational Diagnosis and Evaluation) was a mix of detection and intervention to identify and reduce risky behaviors [11]. The “Proceed” (Policy, Regulatory, and Organizational Constructs in Educational and Environmental Development) in contrast, was characterized by the evaluation on the behavioral determinants post intervention [11]. In the context of screening for infectious diseases, the systematic review of waterborne enteric pathogens by Brouwer, Masters [12] revealed 93 studies were focusing on quantitative microbial risk assessment (QMRA) and 71 studies were developing or validating infectious disease transmission modelling (IDTM). Their review insightfully demonstrated the discrepancies between the high-income countries against the lower- and middle-income countries to demonstrate the roles of screening that considered human ecology, psychology, education, and policymaking. All these models and approaches to risk assessment for public health screening and surveillance were rich and highly useful in their contexts. However, in our context with the urgency to develop and validate mass COVID-19 screening that must also be conducted daily, these approaches to risk assessment had limited direct applications.

Consequently, these gaps in practice and in the literature drove our motivation to innovate a new approach to risk assessment via reliable experts’ judgement. Thus, determining the extent of experts’ agreement on risk screening category based on all possible combinations of predictors emerged as the main objective of this study. To achieve the primary objective, this study would also be addressing the following secondary objectives; (1) to identify and to validate all possible combinations of key predictors, and (2) to develop and to validate an efficient combinatorial-based rating via dashboard-based design with novel algorithm-derived deduction. Fulfilling these objectives would support a new and practical approach to public health screening and surveillance that brought a unique contribution to the literature evidence on approaches to risk assessment. 

## 2. Materials and Methods

### 2.1. Planning

#### 2.1.1. The Conceptual Framework

Figure 1 illustrates the conceptual framework for the three-stages risk prediction modelling based on reliable experts’ ratings on all possible combinations of predictors. This writing focuses on stage 2, which was the key innovation of this novel approach. 

#### 2.1.2. Material

In this study, we opted for Microsoft Excel (Excel 2019, Microsoft Corporation, Washington, DC, USA) as the preferred platform to build a validated instrument for data collection. There were a few reasons for this decision. First, Excel confers a broad flexibility due to extensive mathematics and computation functions, logic testing, visualization, and more. Meanwhile, equipped with a true computer programming language of visual basic application (VBA), the Excel’s capabilities could be extended further. Second, Excel is widely available to potential participants. There are a variety of Excel versions. Most notably, since September 2018, Excel adopted a new computation engine with a leap in abilities to deliver dynamic arrays functions [13]. The dynamic arrays functions greatly simplify the computation steps, allowing users to custom-define and create their own Excel functions [7,14]. However, during the conduct of this study, the new computation engine was only available to users of Excel 365. Thus, we maximized the applications of advanced nested functions available to Excel 2010 and the later versions to preserve an instrument which was widely available to the potential participants of this study.

The third reason was given by the flexibility of commercial options to quickly transform all Excel computations into a web-based application for easy, robust, and professional data collection [15]. Finally, access to the expertise and experience of Excel via one of its members (MSI) enabled the team of medical researchers to be independent in designing, developing, and validating the novel method.

### 2.2. Development and Validation of the Instrument for Data Collection

#### 2.2.1. Step 1: Identify and Validate the Important Predictors

All predictors from the original CHaSe model were retained. Previously, the selection of predictors in the original CHaSe model began with an extensive search of the literature and real COVID-19 case database during the initial phase of the pandemic [5]. Subsequently, the selection and validation of the important predictors were conducted via principal component analyses, discriminant analyses, and medical experts’ validation from a pilot and a real nationwide screening of the target population [4]. As a result, a total of 11 important predictors (designated with letter-code *a* to *k*) and outcomes of three risk categories were shown in Table 1 and Table 2, respectively. 

#### 2.2.2. Step 2: Identify and Validate All Possible Combinations

##### The Count of All Possible Combinations

Excluding predictor coded *b* and *c,* which were mutually exclusive, the total number of possible combinations of *n* variables taken *r* at a time was given by the following equation shown in Formula (1) [16].
(1)$$\sum_{r=0}^{n=9}\frac{n!}{r!\left(n-r\right)!}$$

A total of 512 unique combinations was possible for n=9. The addition of predictor *b* to the 512 combinations yielded another unique 512 combinations. A similar count was applied to predictor *c*, deriving 512+512+512=1536 possible combinations

##### Identify All Possible Combinations of 11 Predictors

The equation as shown in Formula (2) below was inserted into cell A1 of Excel and subsequently extended to cell K2048 [17].
(2)A1=MOD(QUOTIENT(ROW()−1,2^(COLUMN()−1)),2)

The equation returned the modulus (0 or 1) when an integer was divided by 2. The integer meanwhile corresponded to the integer outcome between the dynamic numerator and denominator. The formula derived 2048 unique bits (a combination of 0 and 1) for each table row.

##### Proof of Unique Combinations

In cell M1**,** we inserted nested logic testing with expanding range as shown in Formula (3) below to identify any duplicate.
(3)M1=IF(COUNTIF($L$1:L1,L1)>1,"Tag","")

The function returned empty, proving the final 2048 unique combinations. Figure 2 illustrates this operation.

##### Refining for the Mutually Exclusive Predictors

The table of 2048 unique combinations was transferred hard-coded into a new sheet. Next, the numeric IDs were turned into letter-code IDs by using helper columns using multiple steps operations of excel formulas as shown in Figure 3. Then the 512 combinations that contained the mutually exclusive predictors were identified and removed from the table as shown in Figure 4. The final operations revealed the table containing the validated 1536 possible combinations of 11 key COVID-19 predictors. 

#### 2.2.3. Step 3: Develop and Validate the Dashboard-Based Algorithm-Assisted Experts’ Rating

##### Dashboard-Based Rating

A rating dashboard was designed in a dedicated Excel file to facilitate an efficient collection of experts’ ratings of all 1536 combinations. The rating dashboard was a dynamic display of all possible combinations of predictors run by the Excel computing engine using advanced lookup functions, conditional formatting, and VBA codes. Figure 5 illustrates the link between the table of unique combinations and the dynamic display of the dashboard.

The rating dashboard also employed VBA codes on command buttons for experts to indicate rating judgement and move to the following combination in the sequence. The VBA recorded experts’ ratings in a secured table on a separate worksheet which was hidden and protected. Figure 6 shows the final layout of the interactive dashboard-based rating.

##### The Algorithm-Derived Deduction of Rating

There were two rules for the deduction of risk in the COVID-19 screening.

All predictors signified an additional risk of getting COVID-19 infection. None of the predictors were protective or negatively associated with the risk of getting COVID-19 infection.Therefore, whenever an expert judged a minimum combination of predictors as a high-risk category, all other combinations with a higher count of predictors that contain this minimum combination must also be categorized as a high-risk category. For example, suppose a combination of close contact and fever was rated as high risk. In that case, all other combinations which contain these two predictors must also be rated as high-risk.

As a result, we created a novel algorithm based on advanced combinations of VBA and complicated nested functions and formulas of Excel. Initially, experts needed to rate the risk category for all 11 individual predictors. From then, the algorithm would automatically identify and assign all subsequent combinations with the same combination as the high-risk category based on the above logic operations. Combinations which had been marked by the algorithm would also not be displayed by the dashboard. Thus, the algorithm improved the practicality of the dashboard-based rating by eliminating the need for experts to judge all combinations that could readily be deduced based on prior ratings. The dashboard displayed the sequence of combinations from combinations with the least count of predictors to the largest for ratings’ efficiency. 

Consequently, the algorithm crucially enabled complete data collection of experts’ judgement of risk category for all possible combinations of predictors without having the experts to rate all combinations. Moreover, the innovation resulted in practical data collection with complete information from the labelled dataset for further statistical agreement analyses and prediction models via supervised machine learning.

##### Validation of the Dashboard-Based Algorithm-Assisted Rating

Since this is a novel algorithm, concurrent validation was conducted by taking manual inspection as the gold standard. A pilot rating by one of the researchers was conducted for the manual comparison inspection. For each click of the command button, the rating was manually recorded on a separate excel table in a different file (the manual record) and subsequently compared with the record from the rating dashboard. 

The record of the rating dashboard comprised two lists. The first was the manual rating recorded by VBA when an expert rater clicked the command button to indicate judgement of risk category for the displayed combination of predictors. The second was the auto-rating derived from the algorithm logic deduction described above. 

Validation of both rating dashboards’ lists was substantiated via comparison and inspection with the manual record. Figure 7 demonstrates a section of the rating table that underwent manual validation as described in this step.

### 2.3. Data Collection

#### 2.3.1. Recruitment

All registered medical specialists who were managing the COVID-19 screening either as clinical practitioners, administrators, or policy makers were eligible to participate in this cross-sectional study. We excluded medical experts who were not familiar with basic computer skills including the use of Microsoft Excel. Eligible experts were recruited through simple random sampling. The sampling frame was derived from combining all contacts accessible by all members of the CHaSe medical team. The randomly selected medical experts were invited to participate via a phone call. The number of experts (r) was estimated via the coefficient of variation (cv) among experts by the equation given in Formula (4) [18] (pp. 160–161).
(4)r=2cv

In this study, we accepted cv of 20% (10 expert raters) because of three reasons. First, based on minimum five raters in inter-rater agreement study for instrument content validation [19], cv of 20% provided a higher precision measure but not too high that could be considered excessive. Second, since there was no prior similar study to ours, referencing precision measures from other well-established fields was justified. Finally, the equation was very conservative, where rating variation attributable to the standard error in the actual study may likely be smaller [18]. A total of 13 experts were invited to participate to account for a potential 20% dropout. 

#### 2.3.2. Face Validation and Response Process

Since this was a new method, each expert was individually explained and demonstrated the operation of the rating dashboard and the designated meaning of each risk category. In addition, the researchers verified expert raters’ understanding via online interactive, hands-on practices of rating using the dashboard and ample opportunities to raise any questions or concerns throughout the session. 

Upon completing the rating for all 1536 combinations, each medical expert submitted the completed Excel file to researchers by email. Upon receiving the completed rating dashboard, the rating results from each file were extracted and combined into one master rating table for further analyses.

### 2.4. Analyses

#### 2.4.1. Chance-Corrected Agreement Coefficient (CAC)

Where Pa is the percentage of observed agreement and Pe is the percentage of agreement which could happen by chance, Formula (5) demonstrates the equation for an agreement coefficient that had been corrected for chance agreement [20].
(5)(Pa−Pe)(1−Pe)

In this study, Pe follows the equation by Gwet [18]. Where $P_{k+}$ and $P_{+k}$ are the relative number of items being judged as $x_{k}$ by rater 1 and rater 2, while $\theta_{1}$ and $\theta_{2}$ are the relative number of items that rater 1 and rater 2 have scored respectively, Formula (6) shows the equation for chance-correction used in agreement coefficient 2 (AC_2_) adopted in this study.
(6)[Pk+θ1]+[P+kθ2]2

Compared with the conventional coefficient, the selected AC_2_ has been proven to be relatively resistant to paradox bias both in simulation and empirical studies [21,22]. Additionally, ordinal weight was also selected to represent the partial agreement based on the ranking of the three risk categories employed in this study. 

#### 2.4.2. Inferential Statistics

We adopted statistical benchmarking via cumulative probability based on the scale by Landis and Koch [23]) as proposed by Gwet [18] to interpret the extent of agreement represented by the ordinal-weighted AC_2_. Accordingly, the highest scale, which first reached a minimum of 95% cumulative probability, was accepted to represent benchmarked extent of agreement generalized to the referenced population of experts. The statement of hypotheses was as below:

**H_0_:** 
*There is no substantial agreement among medical experts on the COVID-19 risk categories for all possible combinations of CHaSe predictors.*


H_0_ would be rejected if the cumulative probability of substantial scale reached a minimum of 95%, indicating a less than 5% of chance for the ordinal weighted AC_2_ to fall within this benchmarked class of agreement. 

#### 2.4.3. Statistical Software

Data analyses for the ordinal weighted AC_2_ with error margin and statistical benchmarking were computed via Agreestat software [24].

### 2.5. Ethical Approval

This study was approved by the UniSZA Human Research Ethics Committee (UHREC) with approval ID UniSZA.C/2/UHREC/628-2 Jld 2 (74).

## 3. Results

A total of 11 experts completed the rating (84.6% participation rate). Table 3 summarized the descriptive statistics of counts of ratings by an individual expert. Consistent with our findings from the pilot study, there were noticeable variations of judgement among medical experts. Table 4 meanwhile showed that on average, an expert only needed to provide ratings on 368 out of 1536 possible combinations (24.0%), while the remaining 1168 combinations (76.0%) could readily be rated by the algorithm based on the logic deduction as described above.

On the other hand, Table 5 showed significant *p*-value and the ordinal weighted AC_2_ reached a substantial class of statistical benchmarking. Thus, the null hypothesis was rejected. These results demonstrated that there was a substantial agreement among the medical experts to judge risk categories of the COVID-19 screening based on all possible combinations of 11 predictors in the CHaSe model.

## 4. Discussion

### 4.1. Novel Contribution

To the best of our knowledge, this study is the first to report a method to determine the extent of agreement among the medical experts on risk assessment categories based on all possible combinations of predictors for public health screening and surveillance. This new method allowed inferential statistics that showed a substantial agreement among the medical experts, which vindicate reliable experts’ judgement. The substantial agreement together with complete data of rating among all participating experts for all possible combinations from the study consequently enabled researchers to synthesize a valid labelled training dataset for a supervised machine learning prediction model for the COVID-19 screening and surveillance. Thus, the ability to statistically establish the extent of agreement for all possible combinations of predictors was one of the key contributions of the new method.

Moreover, the new method was also practical to conduct. From the 1536 possible combinations, Table 4 showed that, on average, experts needed to provide ratings for less than a quarter. Yet, they were enough for the novel algorithm to logically deduce high-risk categories for the remaining. On the other hand, from the researcher’s perspective, the dashboard-based rating design removed the tedious tasks of creating 1536 independent questions for the traditional questionnaire-type data collection. Furthermore, answering 1536 questions for each possible combination would likely prove demanding among the expert raters. Even if these two were laboriously addressed, the volume of manual work required to create the questionnaire would have increased the risk of error, compromising the assurance of the content validity of the questionnaire. Therefore, the innovation addressed the fundamental gaps of practice that demand both; a complete and valid combinatorial-based rating and being practical.

The practicality signified repeatability. Since the conduct of this study, the university had endorsed two updates of the COVID-19 risk prediction for screening and surveillance among all staff and students. These updates included adding new important predictors such as anosmia and agnosia [25], refining the focus of prediction on epidemiological links and symptoms, as well as varying the level of stringency of the model following the introduction of safe and effective vaccines [26,27]. Hence, the practicality and repeatability of the new method enabled a public health screening and surveillance that could be shown to be robust, valid, reliable, and responsive to the evolution of our understanding based on the emerging evidence of the pandemic and the consequent changes in national and local policies.

### 4.2. Comparison with the Existing Methods

Existing methods to determine experts’ judgement on combinations of criteria include the use of a panel of experts [28], an open forum of discussions with consensus statements [29], and a systematic approach to reaching consensus such as the Delphi method [30]. In comparison, the novel method adopted in this study offered a practical alternative to survey independent individual experts’ judgement. Independence of judgement removed potential bias from direct influence among the experts attributable to unequal positions, seniority, knowledge, and experience. Furthermore, the independence of judgement also assured no potential bias from vociferous character or other factors which may result in certain individuals exerting dominance or steering discussions toward a certain direction of engagement. These biases could compromise the validity and reliability of the decisions based on combinations of criteria. The experts’ rating employed in this study was free from these biases. As a result, the extent of agreement on combinations of these criteria can be statistically substantiated and generalized to the population of field experts. 

### 4.3. Potential Applications

The ability to statistically generalize a reliable experts’ agreement on combinations of criteria may prove essential. In medical and healthcare practice, the statistical certainty of reliable experts’ judgement for combinations of criteria may serve as the evidence-based decision for policies and practices. These include high-stakes decisions for numerous clinical presentations, escalation of treatment, and urgent resource allocation for time-critical management of various emergencies and critical care. Similarly, in the educational field, a high-stakes examination commonly consists of different purposes, types, and tools of assessments [31]. Although extensive standard-setting methods have been established to determine the passing mark of each assessment, there is no recognized method to determine the extent of agreement on how combinations of these assessments should be judged for the passing criteria of the overall examination. The existing method of either compensatory approach, conjunctive approach, or varying degree of the mixed approach of both is relatively arbitrary and remains a challenge for educational practices [32]. In comparison, the novel combinatorial-based rating reported in this study may be adapted to determine how various combinations of grades of assessment could be judged by experts for the passing criteria of the overall examination. Therefore, establishing statistical validity on the extent of agreement for various combinations of criteria may prove vital in any field that relies on experts’ judgement for high-stakes decision-making.

### 4.4. Chance Correction for Combinatorial-Based Rating

Agreement analyses conducted in this study involved ratings from algorithm-derived deduction which had also been corrected for chance agreement. The notion of chance-corrected agreement via marginal probability was proposed by Cohen [20] to account for an agreement that could happen purely due to chance. However, his idea assumed a distinct entity of the subject being rated and did not account for the algorithm-derived logic deduction for combinations of subjects. Hence, the notion of chance correction for the combinatorial-based rating with or without algorithm-derived logic deduction may be subjected to further research to deduce the most appropriate statistical method for chance correction and how the algorithm-derived deduction may have affected the intra-rater reliability.

Nonetheless, despite the unestablished approach to chance correction for combinatorial-based rating, we justify AC_2_ as the best CAC for this study for three reasons. First, assigning chance-correction probability to auto-rated combinations will only underestimate, rather than overestimate, the extent of agreement. Second, the paradox-resistant AC_2_ gives a less pronounced underestimation of agreement than the traditional CAC, which compute chance correction via marginal probability. In the case of skewed agreement toward one class of rating, the marginal probability computations result in a crucial overcorrection of chance agreement. In this situation, the paradox bias emerges due to a paradoxically low coefficient despite a high percentage of observed agreement. Finally, the results of this study, as shown in Table 3, indeed reported skewed ratings toward high-risk categories for most of the combinations. Thus, we accept the paradox-resistant AC_2_ as the most appropriate coefficient for this study compared to any other paradox-susceptible coefficients such as Krippendorf’s alpha and Fleiss Kappa. 

### 4.5. Limitations

Despite the dashboard-based rating design and the efficient algorithm-derived deduction, there was still a limit on the practicality of developing a model of public health screening and surveillance based on risk assessment on all combinations of key criteria. The equation $2^{n}$ summarises the total possible combinations of n independent predictors. For example, the selection of 13 criteria revealed a total of 8192 possible combinations. Thus, there was an exponential increase in the count of combinations with each count of predictor being added for combinatorial-based rating. As a result, we recommended thorough and valid selections of key criteria must first be established before surveying the field experts for combinatorial-based agreement analyses. Based on the convenience of the dashboard-based rating design and the efficiency of the algorithm-derived deduction, we believed the selections of key criteria should be limited to 12 at most to avoid an excessive cognitive demand among the expert raters.

Additionally, as with any novel data collection method, the expert raters were unfamiliar with the dashboard-based rating design. In this study, a brief session of explanation and hands-on practices were needed to verify the potential participants’ understanding. Thus, an additional step indicating an additional task was required to substantiate the response process and face validation. It also signified that anonymous recruitment was not possible in this study. Nonetheless, from our experience in this study, most experts were able to familiarize themselves with the rating task relatively quickly and were only requiring a few combinations to rate as training during the hands-on practices. Thus, despite these limitations, we believed conducting a larger scale of a combinatorial-based survey among the experts for a more robust and higher precision of agreement analyses would have still been practicable. Furthermore, a more expansive application of this method over time could also address these limitations when more population of raters become familiar with the dashboard-based instrument for combinatorial-based rating. 

### 4.6. Recommendations

Consequently, we invite more studies in public health, wider areas of medicine, and other fields to adopt reliable experts’ judgement as the source of validity for a broader evidence-informed decision-making. In particular, the new approach may have provided a practical and novel approach to developing clinical prediction rules based on the experts’ agreement on various combinations of clinical, radiological, and laboratory criteria for complex clinical presentations. The feasibility and repeatability of the method enable the development of clinical prediction rules which are responsive to the evolution of experts’ understanding of the pathophysiology, complex clinical presentations, and the emergence of new empirical evidence. Furthermore, in certain situations where clinical decision-making is mainly relying on the opinions of medical experts, the statistics of medical experts’ agreement on all possible combinations of important criteria may likely serve as a higher hierarchy of evidence. Therefore, we believe adopting reliable experts’ judgement on various decisions for policy and practice may command wider and better acceptability among various stakeholders.

## 5. Conclusions

The magnitude of the impact of COVID-19 during the initial phase of the pandemic illustrated the critical roles of screening and surveillance for public well-being and safety. In this study, we reported a novel approach for developing and validating COVID-19 screening and surveillance via medical experts’ agreement on risk assessment based on all possible combinations of key predictors. The inferential statistics of agreement analyses revealed a substantial agreement which vindicated reliable experts’ judgement. The reliable judgement and complete rating data from all participating experts for all possible combinations could subsequently be used by researchers as a valid training dataset for the supervised machine learning risk prediction model. The robust, valid, reliable, practical, and efficient novel approach may prove essential for wider public health screening and surveillance of communicable and non-communicable diseases, as well as other medical and non-medical fields.

## Figures and Tables

**Figure 1 ijerph-19-16601-f001:**
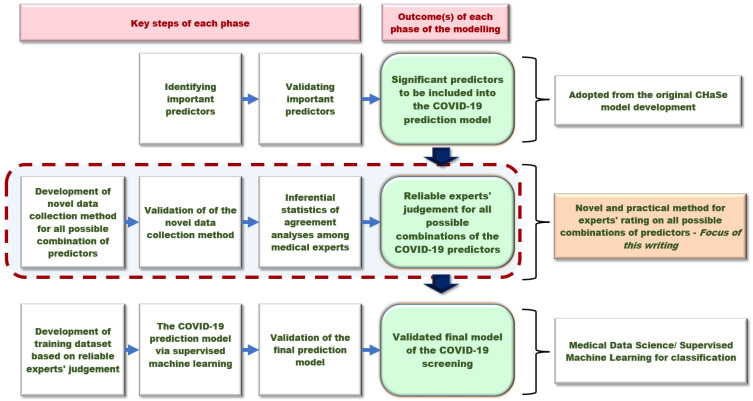
Conceptual framework for the overall methodology of a combinatorial-based prediction model for the COVID-19 Public Health Screening and Surveillance.

**Figure 2 ijerph-19-16601-f002:**
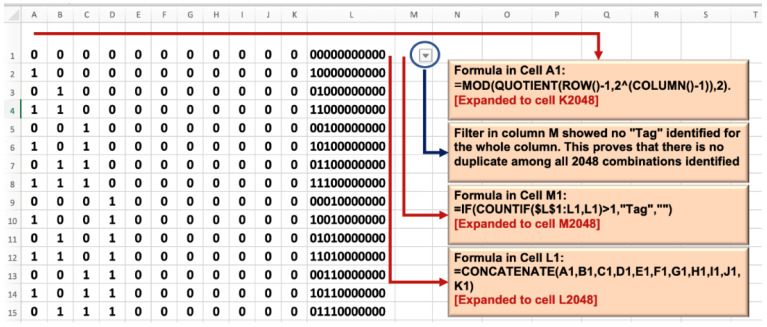
Determining and proving 2048 unique combinations of 11 predictors.

**Figure 3 ijerph-19-16601-f003:**
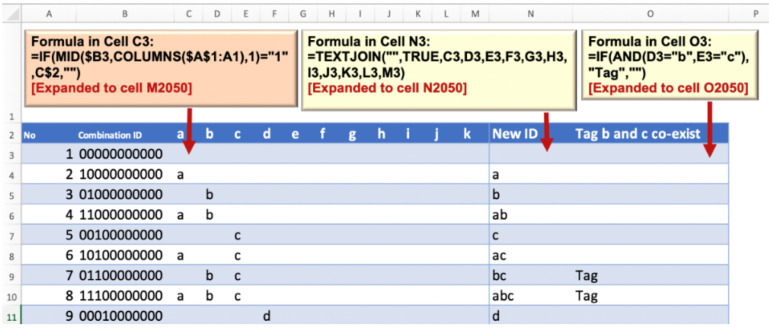
Turning unique bits into unique combinations of alphabetical code of predictors.

**Figure 4 ijerph-19-16601-f004:**
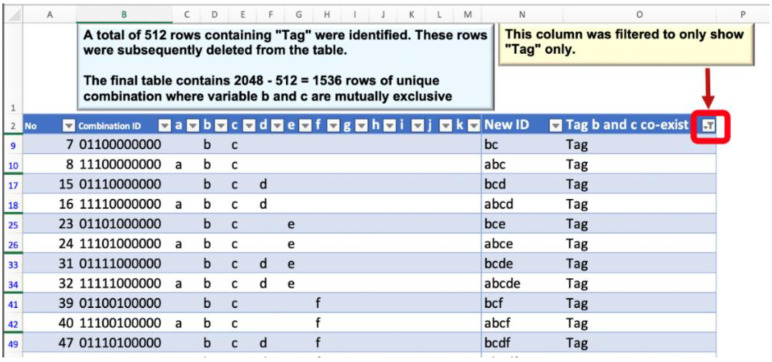
Removing the combinations which contain the mutually exclusive predictors.

**Figure 5 ijerph-19-16601-f005:**
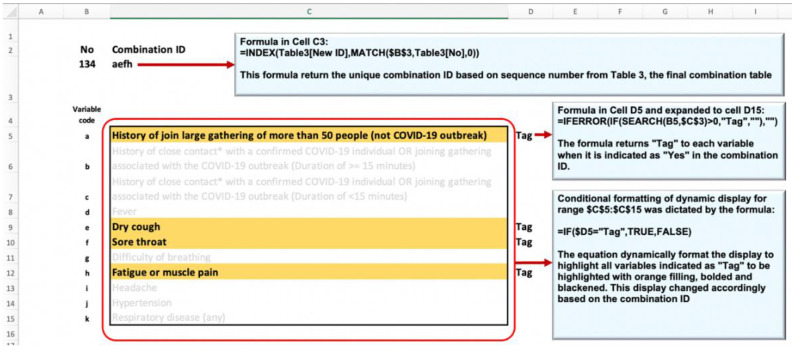
Linking the table of unique combinations with the dynamic display of the rating dashboard.

**Figure 6 ijerph-19-16601-f006:**
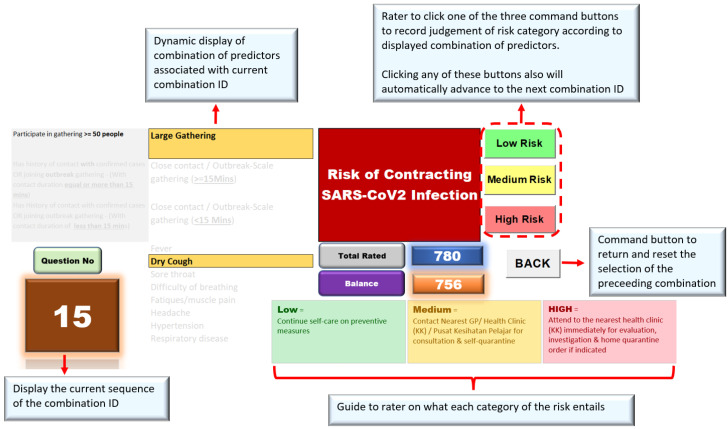
The final layout of the dashboard-based rating.

**Figure 7 ijerph-19-16601-f007:**
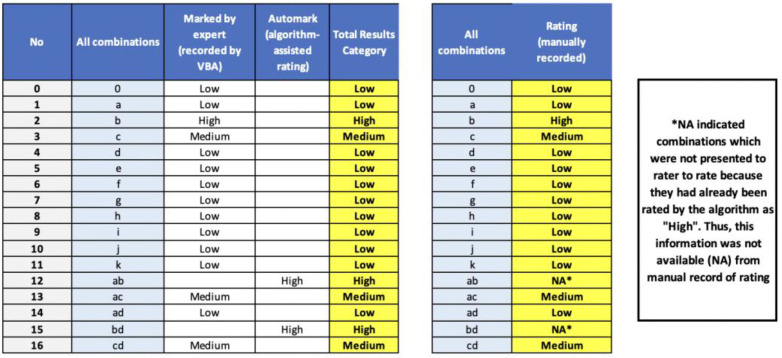
Validation of a section of the dashboard-based rating via comparison and inspection with the manual record.

**Table 1 ijerph-19-16601-t001:** Important predictors of the COVID-19.

No.	Variables	Category	Code
	** *For the last 14 days, the history of;* **	** *Epidemiological Link* **	
1	joining a large gathering of more than 20 people (not COVID-19 outbreak)		a
2	close contact * with a confirmed COVID-19 individual OR joining a gathering associated with the COVID-19 outbreak (Duration of either event ≥ 15 min)		b ^✦^
3	close contact * with a confirmed COVID-19 individual OR joining a gathering associated with the COVID-19 outbreak (Duration of either event < 15 min)		c ^✦^
	** *Currently experiencing;* **	** *Symptoms* **	
4	fever		d
5	dry cough		e
6	sore throat		f
7	difficulty of breathing		g
8	fatigue or muscle pain		h
9	headache		i
	** *Existing medical conditions;* **	** *Comorbidities* **	
10	hypertension		j
11	respiratory disease (any)		k

* Close contact was defined as either working together in the same place (e.g., office, room, working station), living together in the same house, traveling together in the same vehicle, or being in the same assembly or space (for example class, meeting, course, occasion). ^✦^
*Predictors coded as b and c were mutually exclusive*.

**Table 2 ijerph-19-16601-t002:** Advice based on risk categories of the COVID-19 screening.

Risk Category	Advice
Low	You are advised to practice preventive measures to avoid COVID-19 infection.
Medium	You are required to contact the UniSZA health center for further evaluation before coming to campus. You are advised to practice preventive measures to avoid COVID-19 transmission to others.
High *	You are required to stay at home and immediately contact the UniSZA health center or attend a government health clinic for further evaluation, treatment, and COVID-19 test if necessary. You are advised to practice preventive measures to avoid COVID-19 transmission to others.

* respondents in this category would also be contacted by UniSZA health center as part of active case detection if they did not initiate contact.

**Table 3 ijerph-19-16601-t003:** Summary of class of rating by expert rater.

	Rating Type *	Low	Medium	High	Total Rating
Rater 1	Manual rating	39	250	10	1536
	Auto-rating	-	-	1237	
Rater 2	Manual rating	205	65	25	1536
	Auto-rating	-	-	1241	
Rater 3	Manual rating	40	101	16	1536
	Auto-rating	-	-	1379	
Rater 4	Manual rating	78	434	2	1536
	Auto-rating	-	-	1022	
Rater 5	Manual rating	518	250	2	1536
	Auto-rating	-	-	766	
Rater 6	Manual rating	217	115	67	1536
	Auto-rating	-	-	1137	
Rater 7	Manual rating	81	82	31	1536
	Auto-rating	-	-	1342	
Rater 8	Manual rating	244	130	56	1536
	Auto-rating	-	-	1106	
Rater 9	Manual rating	123	154	39	1536
	Auto-rating	-	-	1220	
Rater 10	Manual rating	130	270	7	1536
	Auto-rating	-	-	1129	
Rater 11	Manual rating	165	96	10	1536
	Auto-rating	-	-	1265	

* Auto-rating was algorithm-derived logic deduction on high-risk category based on the manual ratings by the experts.

**Table 4 ijerph-19-16601-t004:** Summary of type of rating.

Count of Rating	Sum (11 Raters)	Average (%)
Manual rating	4052	368.36 (24.0%)
Auto-rating	12,844	1167.64 (76.0%)
Total	16,896	1536.00 (100.0%)

**Table 5 ijerph-19-16601-t005:** Inferential statistics of agreement analyses.

		Inference		Cumulative Probability on Koch and Landis Scale						Statistical Benchmarking
		95% CI	*p*-Value	Almost Perfect	Substantial	Moderate	Fair	Slight	Poor	
**Unweighted AC_1_**	0.72	0.70 to 0.74	<0.001	0.03	0.97	1.00	1.00	1.00	1.00	Substantial
**Ordinal Weighted AC_2_**	0.81	0.79 to 0.82	<0.001	0.49	1.00	1.00	1.00	1.00	1.00	Substantial

## Data Availability

Not applicable.

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
