# Peer review of "Medical Experts’ Agreement on Risk Assessment Based on All Possible Combinations of the COVID-19 Predictors—A Novel Approach for Public Health Screening and Surveillance"

_ijerph, 2022, doi:10.3390/ijerph192416601_

Round 1
Reviewer 1 Report
This is an interesting and well prepared manuscript.
There are only a few points that I would ask the authors to consider:
Please clearly mention the objectives of the study. I read the introduction but could not determine the particular objectives and idea of this study in the last paragraph of the introduction. Please also clearly mention the contribution of this study to the literature.
The figure1 isn’t clear
Line 132 equation =512 is not standard
Line 138 equation is not standard
Line 146 equation is not standard
Line 267 equation is not standard
Line 270 equation is not standard
Line 299 I read the section Result, but could not get more information in there.
Reviewer 2 Report
• The flow of the paper is well formatted and formed to understand the key points discussed.
• In Abstract, kindly mention which method is utilized for 1536 possible combinations evaluation with 11 key predictors of COVID-19 (as per the section 2.2.2 in article).
• In Section 2, Materials and Methods, the appropriate discussion on Materials is not provided. Kindly check and update.
• The methodology is explained in detail and all methods are elaborated well.
• Kindly check for the quality of figure 6.
• Kindly check the formatting of headings and sub-heading. Uniformity is missing.
• In section 2.2.2, the lines mentioned “The algorithm-derived deduction of rating” and “Validation of the dashboard-based algorithm-assisted rating” are sub-heading under this sub-section or are they sub-sections. Kindly change according to format.
• Results and discussion in section 3 and 4 are presented well.
Round 2
Reviewer 1 Report
Good job!
Reviewer 2 Report
The queries raised in earlier version has been addressed upto my satisfaction.